# Curiosities above $c = 24$

**A. Ramesh Chandra**[1][*] **and Sunil Mukhi**[1]

**1** Indian Institute of Science Education and Research,
Homi Bhabha Rd, Pashan, Pune 411 008, India

[*] ramesh.ammanamanchi@gmail.com

## Abstract

Two-dimensional rational CFT are characterised by an integer $\ell$, related to the number of zeroes of the Wronskian of the characters. For two-character RCFT's with $\ell < 6$ there is a finite number of theories and most of these are classified. Recently it has been shown that for $\ell \geq 6$, there are infinitely many admissible characters that could potentially describe CFT's. In this note we examine the $\ell = 6$ case, whose central charges lie between 24 and 32, and propose a classification method based on cosets of meromorphic CFT's. We illustrate the method using theories on Kervaire lattices with complete root systems. In the process we construct the first known two-character RCFT's beyond $\ell = 2$.



# 1 Introduction and Review

The organisation of rational conformal field theories by their number of characters is interesting for both physics and mathematics. A classification method for such theories was first proposed in [1] and much progress has been made in recent years, both in the physics [2–5] and mathematics [6–10] literature. This method is based on the following facts: the partition function $Z(\tau, \bar{\tau})$ of a CFT is modular-invariant but not holomorphic, while the characters $\chi_i(\tau)$ are holomorphic (except at infinity) but not modular-invariant – they transform as vector-valued modular functions. However there is a modular linear differential equation (MLDE) satisfied by the characters, that is both holomorphic and modular invariant. This is highly constraining and can be used to classify those MLDE that give rise to "admissible" characters – holomorphic vector-valued modular functions of $q = e^{2\pi i \tau}$ that have non-negative integer coefficients in their $q$-expansion.

From the physical point of view, the number of characters (minus 1) is equal to the number of non-trivial critical exponents, so if we want to study critical systems with just one or two exponents then we can focus on two- or three-character theories and study all of them together. One can, for example, identify classes of theories that have no marginal deformations – these are sometimes called "perfect metals" [11] and examples can be found in [3, 4]. In contrast, traditional classifications through minimal series of extended chiral algebras [12, 13] have rapidly growing numbers of characters and most often, only the first few members of the series are physically interesting.

From the mathematical point of view, single-character ("meromorphic") theories are important due to their close relation to even, self-dual lattices and automorphic forms. They necessarily have central charge $c = 8n$ for integer $n$. The case of $c = 24$ is particularly interesting because here there are infinitely many admissible characters but only 71 RCFT's [14]. Of these, 24 are related to even, self-dual lattices while the remaining are extensions involving orbifolding and other field-theoretic constructions. One such CFT, the celebrated Monster Module, conjectured to be the unique meromorphic CFT at $c = 24$ without currents, has as its automorphism group the Monster – the largest simple sporadic finite group. With more than one character, we have vector-valued modular functions. These are less well-studied in the mathematical literature but there is a growing body of work based on MLDE in the case of two and three characters [6–10].

Let us very briefly review classification method and some new results which will be relevant for our discussion. A detailed review with several new results can be found in [15]. The characters being vector-valued modular function can be written as the independent solutions of a modular invariant differential equation. Such a differential equation can be built using the modular covariant differential operator $D = \frac{1}{2\pi i} \partial_\tau - \frac{k}{12} E_2(\tau)$. Here, $k$ is the weight of the object on which the operator is acting and $E_2(\tau)$ is the second Eisenstein series. Note that this differential operator augments the weight by 2. In the case of two-characters (which will be the main focus for the rest of the paper), the MLDE takes the form

$$\left(D^2 + \phi_2(\tau)D + \phi_4(\tau)\right)\chi = 0. \tag{1}$$

For the above equation to be modular, we see that $\phi_2$ and $\phi_4$ must transform with weights 2 and 4 respectively. These functions can be given in terms of the Wronskians of the MLDE as follows

$$\phi_2 = -\frac{W_1}{W}, \quad \phi_4 = \frac{W_0}{W},$$

$$W_0 = \begin{vmatrix} D\chi_0 & D\chi_1 \\ D^2\chi_0 & D^2\chi_1 \end{vmatrix}, \quad W_1 = \begin{vmatrix} \chi_0 & \chi_1 \\ D^2\chi_0 & D^2\chi_1 \end{vmatrix}, \quad W = \begin{vmatrix} \chi_0 & \chi_1 \\ D\chi_0 & D\chi_1 \end{vmatrix}. \tag{2}$$

The functions $\phi_2$ and $\phi_4$ need not be holomorphic, but can be any rational combination of $E_4(\tau)$ and $E_6(\tau)$ of the correct weight. Since the characters themselves are holomorphic, we see from the above equations that the only singularities of $\phi_2$ and $\phi_4$ arise from the zeroes of $W$. The nature of solutions of the MLDE depends significantly on the singular behaviour of these functions, which is in turn determined by the zeroes of $W$. Let the number of zeroes be equal to $\frac{\ell}{6}$ with $\ell$ taking the values $0, 2, 3, 4, \cdots$. The fractional nature of zeroes is due to the presence of two orbifold points $\tau = i, e^{\frac{2\pi i}{3}}$ in the torus moduli space where the functions are allowed to vanish with degrees $\frac{1}{2}$ and $\frac{1}{3}$ respectively. This parameter $\ell$ satisfies an important relation with the central charge and conformal dimensions, coming from the Riemann-Roch theorem [1]:

$$\alpha_0 + \alpha_1 + \frac{\ell}{6} = -\frac{c}{12} + h + \frac{\ell}{6} = \frac{1}{6}, \tag{3}$$

where $\alpha_0 = -\frac{c}{24}, \alpha_1 = -\frac{c}{24} + h$. To classify a theory one fixes a value of $\ell$, writes the most general MLDE for this $\ell$. The value of $\ell$ determines the form of $\phi_2$ and $\phi_4$. Since these are rational combinations of the standard Eisenstein series, they will contain a finite number of arbitrary parameters. One then solves the $q$-series as a function of the parameters of the equation, and tunes the parameters so that the $q$-series has integer coefficients upto a high order. There are various checks to ensure that the candidate characters found in this way are genuinely admissible (i.e. they have non-negative integral coefficients to all orders). Finally, one tries to reconstruct the CFT corresponding to these characters, if it exists [16].

For two-character theories, $\ell$ is known to be even. For $\ell = 0$ there is one free parameter in the MLDE, and it was shown in [1] that there are precisely 10 values of it such that the solutions are admissible as characters. Each of them was subsequently identified (with some caveats relating to non-unitarity and/or degeneracy of the vacuum[1]) with a CFT of a definite central charge in the range $0 < c \leq 8$, most of which are level-1 Wess-Zumino-Witten models that contain the integrable primaries of the corresponding Kac-Moody algebra. The corresponding Lie algebras belong to the Deligne series [17].

For $\ell = 2$, the analysis of [2,3,18] showed that again the MLDE has a single parameter and there are admissible characters for precisely 10 values, corresponding to central charges in the range $16 \leq c < 24$. Most of these theories have a Kac-Moody algebra but do not correspond to WZW models. Their characters can be understood as very special combinations of Kac-Moody characters which could not have been easily discovered by searching for them directly. An important step in identifying them as actual RCFT's was taken in [3] where it was shown that they are cosets of meromorphic CFT's at $c = 24$ by the $\ell = 0$ theories. This explains their range of central charges, and also sets up relationships between families of RCFT's with different $\ell$ that uses meromorphic theories with $c = 8n$ as an intermediate step. An important outcome of the above investigations is that for $\ell = 0$ and $2$, every pair of admissible characters actually leads to a CFT, modulo the caveats mentioned above.

The case of $\ell = 4$, discussed in [15] is somewhat enigmatic: there are precisely three admissible pairs of characters but no concrete construction of a corresponding CFT is known so far[2]. The case of $\ell \geq 6$ presents interesting new features. Here, results of [15] (and related, earlier work of Harvey-Wu [5]) show how to construct, for the first time, an infinite number of admissible pairs of characters. In [15] it was proved that this classification is complete for all $\ell$. Below, we will focus on $\ell = 6$ and give an explicit construction in this case. This then paves the way to address the question of which among the infinite set of admissible characters

---

[1]These are carefully explained in [15].

[2]Two of the three $\ell = 4$ character pairs were noted in [19]. In this reference the term "extremal" was used to refer to CFT's with $\ell < 6$. With that definition, we are studying the existence of "non-extremal" two-character CFT in this work, for the first time as far as we know.

correspond to actual RCFT's.

Recall that for the analogous problem with one character, which first arises at $c = 24$ [14], a straightforward family of CFT's can be constructed using free boson theories on even self-dual lattices. Then one considers generalisations such as orbifolds of these CFT's and more exotic constructions. In this sense, the problem addressed in the present work may be seen as a two-character analogue of that in [14]. We will follow an approach inspired by that of [3], namely to try and define $\ell = 6$ CFT as cosets of meromorphic theories. Thereafter, for over a hundred examples, we will be able to associate definite RCFT's with given pairs of $\ell = 6$ characters. To our knowledge these are the first known irreducible (i.e. not direct-product) two-character RCFT's beyond $\ell = 2$.

## 2 Admissible characters for $\ell = 6$

We now briefly review how admissible characters for $\ell = 6$ are found. It was proposed in [15] that the natural category of objects to study is not admissible characters, but "quasi-characters" – which have the same holomorphic and modular properties as characters, but are allowed to have negative integer coefficients in their $q$-series. Clearly, admissible characters are a subset of quasi-characters. Some key observations for the case of two characters are:

(i) A complete set of quasi-characters is known for $\ell = 0, 2, 4$,

(ii) Quasi-characters can be added to each other preserving holomorphicity, modularity and integrality, as long as their central charges differ by a multiple of 24,

(iii) Adding quasi-characters augments the $\ell$ value by multiples of 6.

The table below gives the full set of values at which the quasi-characters occur for the $\ell = 0$ MLDE. These are classified by their fusion rule class. There are four fusion classes: LY denotes the fusion class of the Lee-Yang minimal model, $A_1$,$A_2$ and $D_4$ denote the fusion classes of the respective WZW models at level 1. From this data, the quasi-characters themselves are easily constructed from the MLDE.

Table 1: Complete list of quasi-characters for $\ell = 0$

| Class | $c$ | $h$ | $n$ values |
|-------|-----|-----|------------|
| LY | $\frac{2}{5}(6n+1)$ | $\frac{n+1}{5}$ | $n \neq 4 \bmod 5$ |
| $A_1$ | $6n+1$ | $\frac{2n+1}{4}$ | $n \in \mathbb{N}$ |
| $A_2$ | $4n+2$ | $\frac{n+1}{3}$ | $n \neq 2 \bmod 3$ |
| $D_4$ | $12n+4$ | $\frac{2n+1}{2}$ | $n \in \mathbb{N}$ |

To illustrate the points (ii) and (iii) from above, let us consider a simple example. Consider two quasi-character solutions in the $A_1$ class, one at $n = 0$ and the other at $n = 4$. The $n = 0$ case gives us the familiar characters of $A_{1,1}$ WZW model. The $n = 4$ case [15] however has a negative coefficient at level 1, though all of its coefficients are integers. The central charges of the two cases being $c = 1$ and $c = 25$ respectively, differ by 24. A general feature of quasi-characters in Table 1 is that whenever central charges differ by 24, they transform identically under modular transformations. Our $n = 0$ and $n = 4$ cases have the same modular transformations and they can be added without destroying the modularity:

$$\chi_i^{n=4} + N_1 \chi_i^{n=0}. \tag{4}$$

Here $N_1$ is an integer parameter, and $i = 0$ and $i = 1$ represent the vacuum and primary characters of the new solution. Let us determine the $\ell$ value of this new solution given by addition of two $\ell = 0$ quasi-characters. To do this we need to determine the critical exponents of this new solution and use the Riemann-Roch equation given in Eq. (3). Recall that for the $n = 0$ case, $\alpha_0 = -\frac{1}{24}$ and $\alpha_1 = \frac{5}{24}$, and for the $n = 4$ case, we see from Table 1 that $\alpha_0 = -\frac{25}{24}$ and $\alpha_1 = \frac{29}{24}$. The new vacuum character in Eq. (4) starts with $q^{-\frac{25}{24}}$ but the new primary character starts with $q^{\frac{5}{24}}$. Thus the new critical exponents are $\alpha_0 = -\frac{25}{24}$, $\alpha_1 = \frac{5}{24}$ and $c = 25, h = \frac{6}{5}$. In other words, the central charge increased by 24 and the conformal dimension increased by 1. Using Eq. (3), we immediately see that by adding two $\ell = 0$ quasi-characters, we generated a $\ell = 6$ solution. We can tune the integer parameter $N_1$ in Eq. (4) such that the new solution has all positive integer coefficiens, i.e., is admissible. Thus, the addition of quasi-characters is a useful technique to construct admissible characters. In particular, $\ell = 0$ quasi-characters (of Type I, as defined in [15]) generate infinite sets of admissible characters at $\ell = 6m$ for every positive integer $m$.

Section 5.2 of [15] provided a set of some of the admissible characters with $\ell = 6$ that are produced by adding a pair of quasi-characters with $\ell = 0$. The full set of $\ell = 6$ admissible characters obtained as linear combinations of quasi-characters is given here in Table 2, where we display the values of $c$ and the primary dimension $h$, as well as the combination of quasi-characters that they represent. Like quasi-characters, these too are labelled by their fusion rule class.

Notice that only two distinct quasi-characters are being added (if we added a third one, $\ell$ would increase to 12). Also in each case, the second quasi-character in the sum is actually a valid character (lying in the Deligne series), while the first one has a single negative term in its $q$-expansion. As a consequence, the central charge values in this table differ by 24 from those of the Deligne series. The addition leads to a change in the sign of this negative term if the free integer $N_1$ is chosen to be greater than some minimum value in each case[3].

Table 2: $\ell = 6$ pairs obtained by addition of quasi-characters

| No. | $c$ | $h$ | Character sum |
|---|---|---|---|
| 1 | $\frac{122}{5}$ | $\frac{6}{5}$ | $\chi_{LY}^{n=10} + N_1 \chi_{LY}^{n=0}$ |
| 2 | $25$ | $\frac{5}{4}$ | $\chi_{A_1}^{n=4} + N_1 \chi_{A_1}^{n=0}$ |
| 3 | $26$ | $\frac{4}{3}$ | $\chi_{A_2}^{n=6} + N_1 \chi_{A_2}^{n=0}$ |
| 4 | $\frac{134}{5}$ | $\frac{7}{5}$ | $\chi_{LY}^{n=11} + N_1 \chi_{LY}^{n=1}$ |
| 5 | $28$ | $\frac{3}{2}$ | $\chi_{D_4}^{n=2} + N_1 \chi_{D_4}^{n=0}$ |
| 6 | $\frac{146}{5}$ | $\frac{8}{5}$ | $\chi_{LY}^{n=12} + N_1 \chi_{LY}^{n=2}$ |
| 7 | $30$ | $\frac{5}{3}$ | $\chi_{A_2}^{n=7} + N_1 \chi_{A_2}^{n=1}$ |
| 8 | $31$ | $\frac{7}{4}$ | $\chi_{A_1}^{n=5} + N_1 \chi_{A_1}^{n=1}$ |
| 9 | $\frac{158}{5}$ | $\frac{9}{5}$ | $\chi_{LY}^{n=13} + N_1 \chi_{LY}^{n=3}$ |

---

[3]As explained in [15], to get the most general admissible characters one can take $N_1$ to be integer and then change the overall normalisation to get a non-degenerate ground state whenever possible. Alternatively one can take rational linear combinations, chosen so that the sum again has integral coefficients. Here we follow the first approach.

It is amusing that the actual $\ell = 6$ MLDE played no role in constructing Table 2. Our process of adding quasi-characters with $\ell = 0$ automatically augments the value of $\ell$ while preserving the modular transformations and integrality. In view of the general completeness proof in [15], this procedure exhausts all admissible $\ell = 6$ characters. We will find it useful to explicitly exhibit how this completeness operates in the specific class of examples of interest here, namely $\ell = 6$.

The steps in the proof are as follows. The fusion categories for two-character theories are completely classified [20,21], and in [15] we have found, in particular, $\ell = 0$ quasi-characters for every allowed value of the central charge compatible with these fusion rules (see Table 1). But in fact the fusion category classification applies to all values of $\ell$ since it only uses the fact of having two characters. Thus, the allowed central charges for $\ell = 6$ must lie in the same list. Now adding $\ell = 0$ quasi-characters always augments $\ell$ by multiples of 6. Thus, the set of $\ell = 0$ quasi-characters can be thought of as a basis for the characters with any value of $\ell$ that is divisible by 6.

Next, by looking at the $q$-series, it is easily verified that the only way to produce $\ell = 6$ solutions using this basis is to add precisely two quasi-characters – and the values of their central charge must differ by 24. Additionally if the result is to be admissible, then any negative signs in the quasi-characters being added must turn positive after addition. Now suppose the sum is of the form $\chi^{c+24} + N_1 \chi^c$. Let us focus on the negative signs in the individual terms in this sum. Suppose first that $\chi^c$ is admissible, thus it has all non-negative terms and also $0 < c \leq 8$. In that case $\chi^{c+24}$ has a central charge in the range $24 < c \leq 32$. In [15] we have noted that Type I quasi-characters in this range have a single negative sign, which moreover occurs at the first level above the ground state in the identity character, i.e. in the term of order $q^{-\frac{c}{24}}$. In the sum, the leading term of $\chi^c$ contributes precisely to the same power of $q$. Therefore a suitable choice of $N_1$ will make the sum admissible.

Finally, supposing $\chi^c$ is not itself admissible, then both $\chi^c$ and $\chi^{c+24}$ contain negative terms in their $q$-series. One can verify from the $q$-coefficients that no value of $N_1$ will turn all the negative terms positive. Thus, as claimed, the above classification of $\ell = 6$ admissible characters is complete.

As a confirmation, let us note that the MLDE for $\ell = 6$ initially has four free parameters. It can be parametrised as follows:

$$\left( D^2 + \mu_2 \frac{E_4^2 E_6}{E_4^3 + \mu_1 \Delta} D + \frac{(\mu_3 E_4^3 + \mu_4 \Delta) E_4}{E_4^3 + \mu_1 \Delta} \right) \chi = 0 \tag{5}$$

and we see that the coefficient functions have a "movable" pole at $E_4^3 + \mu_1 \Delta = 0$. Clearly the location of this pole is determined by $\mu_1$. Now the Riemann-Roch theorem fixes $\mu_2 = 1$, and $\mu_3$ is determined by the central charge. This leaves the parameters $\mu_1, \mu_4$. Next we require that the solution is not logarithmic around the free pole, which turns out to relate $\mu_1$ and $\mu_4$. That finally leaves one free parameter in addition to the central charge. A sum of the form $\chi^{c+24} + N_1 \chi^c$ also has one free parameter, $N_1$, in addition to the central charge (for purposes of this argument we can treat $N_1$ as a real number rather than an integer, since the sum solves the MLDE for any real $N_1$). Thus the number of parameters in our proposed general solution is equal to the number in the MLDE, consistent with our solutions being complete. Such a parameter count can actually be done for higher values of $\ell$ that are multiples of 6, but we leave that for a future investigation.

The question to which we now turn is, how do we identify some (or all) of these admissible characters with actual CFT's?

# 3 Coset construction for $\ell = 6$ CFT's

In order to identify CFT's for these $\ell = 6$ characters, we will use the novel coset construction first used in [3] to identify $\ell = 2$ CFT's. Let us briefly recall this construction. Say we have a meromorphic theory $\mathcal{H}$ having a Kac-Moody algebra, as well as possible higher-spin chiral algebras. If $\mathcal{D}$ is an affine theory (i.e. a diagonal invariant) of a Kac-Moody algebra which in turn is a direct summand of the algebra of $\mathcal{H}$, then we can construct the coset $\mathcal{C} = \mathcal{H}/\mathcal{D}$ as explained in [3]. The decomposition of the character of $\mathcal{H}$ in terms of the characters of $\mathcal{D}$ determines the characters of $\mathcal{C}$ via the following relation:

$$\chi_0^{\mathcal{H}} = \chi_0^{\mathcal{D}} \cdot \chi_0^{\mathcal{C}} + \sum_{i=1}^{n-1} M_i \, \chi_i^{\mathcal{D}} \cdot \chi_i^{\mathcal{C}}, \tag{6}$$

where the integers $M_i$ are multiplicities. Notice that this bilinear relation is completely holomorphic. From this we immediately have relations among central charges and conformal dimensions: $c^{\mathcal{H}} = c^{\mathcal{D}} + c^{\mathcal{C}}$ and $h_i^{\mathcal{C}} + h_i^{\mathcal{D}} \in \mathbb{N}$.

The central charges, conformal dimensions and $\ell$-values of a coset pair are known [3] to satisfy:

$$\ell + \tilde{\ell} = 2 + \tfrac{1}{2}(c + \tilde{c}) - 6(h + \tilde{h}). \tag{7}$$

From this we see that $\ell = 0$ and $\tilde{\ell} = 6$ characters pair up such that $c + \tilde{c} = 32$ and $h + \tilde{h} = 2$. Moreover from their modular properties we find they satisfy the bilinear relation:

$$\chi_0(\tau)\tilde{\chi}_0(\tau) + M\chi_1(\tau)\tilde{\chi}_1(\tau) = j^{\frac{1}{3}}(\tau)(j(\tau) + \mathcal{N}). \tag{8}$$

The integer M counts the multiplicity with which the non-identity primary occurs. On the RHS, we have the character of a potential $c = 32$ meromorphic theory, which depends on an integer parameter $\mathcal{N}$. From the $q$-expansion of the RHS:

$$j^{\frac{1}{3}}(\tau)(j(\tau) + \mathcal{N}) = q^{-\frac{4}{3}}(1 + (\mathcal{N} + 992)q + \cdots). \tag{9}$$

We see that such a theory, if it exists, has $\mathcal{N} + 992$ spin-1 currents. This imposes a bound $\mathcal{N} \geq -992$. Since the spin-1 currents form a Kac-Moody algebra, which contributes to the central charge via the Sugawara construction, $\mathcal{N}$ cannot be arbitrarily large or else $c$ would exceed 32. The upper bound on $\mathcal{N}$ is achieved when the currents form a $D_{32,1}$ Kac-Moody algebra, for which $\mathcal{N} + 992 = 2016$. Thus, $-992 \leq \mathcal{N} \leq 1024$. Since the $\mathcal{N} + 992$ currents come from the currents of the $\ell = 0$ and $\ell = 6$ characters that form a coset pair, one can relate $\mathcal{N}$ to the integer $N_1$ appearing in Table 2, placing bounds on the latter (the precise bound for $N_1$ will vary by fusion category).

We have established that our $\ell = 6$ characters are cosets of $c = 32$ meromorphic characters by $\ell = 0$ theories. If now we can show that the meromorphic character in question really corresponds to a CFT, then it follows that the $\ell = 6$ characters also describe a genuine CFT. Thus we need to identify $c = 32$ meromorphic theories whose chiral algebra contains any of the Kac-Moody algebras arising in $\ell = 0$ theories as a direct summand.

Meromorphic CFT's with $c = 32$ are far from being classified, unlike the cases of $c = 8, 16$ and 24 where they are completely classified. As noted above, the simplest constructions for such CFT's are based on even unimodular lattices. Such lattices have three important properties in dimensions 8, 16, 24 which were crucial for the classification problem in these dimensions [22–24]:

(i) The root system (set of points of norm 2) of these lattices is either empty, or has rank equal to the dimension of the lattice. Moreover there is a unique lattice for each root system.

(ii) If the root lattice is a sum of several irreducible components, then all the components have the same Coxeter number.

(iii) The number of lattices for dimension $\leq 24$ is small, namely $(1, 2, 24)$ for dimension $(8, 16, 24)$ respectively. This property actually follows from the two above.

Lattices whose root systems have rank equal to the dimension of the lattice are said to have a *complete root system*. Thus, all even unimodular lattices with dimension less than or equal to 24 have a complete root system, except for the Leech lattice which has none.

The above three properties can be translated into properties of meromorphic CFT's with $c \leq 24$. For a lattice CFT, spin-1 currents arise from the roots of the lattice as well as Cartan generators of the form $\partial X^i$ where $i$ runs over the dimension of the lattice. The first property above says that either there are no roots, in which case the abelian currents form a $U(1)^c$ algebra, or there are roots which combine with the Cartan generators to form a semi-simple Kac-Moody algebra (direct sum of non-abelian factors) with a Sugawara central charge $c$. The second case will be referred to as a *complete Kac-Moody algebra* because in this case the structure of the non-abelian algebra (integrable primaries, null vectors etc.) determines the CFT. In the first case the situation is less clear, as the abelian algebra alone does not tell us enough about the theory.

Going beyond lattice theories, the situation becomes more complex. For example, we encounter non-simply-laced factors in the Kac-Moody algebra and the total rank of this algebra can be $< 24$. Nonetheless, we refer to such Kac-Moody algebras as complete if they are semi-simple and their central charge is equal to the total central charge of the theory. With this definition, the only incomplete Kac-Moody algebras at $c = 24$ are the Leech lattice CFT with $U(1)^{24}$ and the Monster CFT, obtained by orbifolding the Leech lattice CFT to remove the 24 abelian currents.

The second property of such lattices listed above, applied to a meromorphic CFT with $c \leq 24$, says that if its Kac-Moody algebra is a direct sum of irreducible components, then the dual Coxeter number $\check{h}$ is the same for each component. If we go beyond lattice CFT's then a more general version of the result holds, namely the ratio of $\check{h}$ to the level $k$ is the same for each of the components [14].

The third property listed above for lattices in dimension $\leq 24$ – that their number is small – is also related to, though does not immediately imply, a comparably small number of meromorphic CFT's with $c \leq 24$. The actual number turns out to be $(1, 2, 71)$ for $c = (8, 16, 24)$.

None of these restrictive properties is applicable once we go above $c = 24$, making the classification there very difficult. To start with, in 32 dimensions the lower bound on the number of even unimodular lattices is itself of order $10^9$, as shown in [25]. Quite contrary to the cases in $\leq 24$ dimensions, the root systems of these 32 dimensional lattices have all possible ranks, ranging from 0 (empty root system), $1, 2, \cdots, 31, 32$ (complete root system). In fact, the vast majority of these lattices have incomplete root systems. If the rank of the root system is $r < 32$, the corresponding CFT has an additional $U(1)^{32-r}$ factor in its spin-1 algebra and by our definition its Kac-Moody algebra is incomplete. If we go beyond lattices and construct more general $c = 32$ meromorphic CFT by methods parallel to those of [14], the total rank of abelian and non-abelian algebras together will typically fall below 32 and one will encounter both complete and incomplete cases.

Things become much more manageable if we start out by restricting ourselves to lattices with complete root systems. There are only 132 (out of more than a billion!) such indecomposable lattices, and they were classified by Kervaire in [26]. There are 119 distinct rank 32 root systems, all simply laced, corresponding to these lattices (unlike in $c \leq 24$, a few inequivalent lattices have the same root system). The simplest examples of $c = 32$ meromorphic CFT's can be now constructed from these 132 lattices, and all of them will have a complete Kac-Moody

algebra at level 1 with rank 32 and a Sugawara central charge equal to 32.

We can now return to our initial problem. We pick any of the above CFT's which have a Kac-Moody algebra containing an $\ell = 0$ affine theory as a direct summand, and take the coset to get an $\ell = 6$ theory. Since all the Kervaire lattices have simply laced root systems, the Kac-Moody algebras only have simply laced Lie components. Hence these cases have very similar properties to the cosets considered in [3][4]. Thus, for a sizable number of 32-dimensional lattices with complete root systems, the coset theory is completely well-defined as a CFT.

The list of $\ell = 6$ CFT's obtained as cosets of these theories is given below in Table 3. The last column links to lists in Appendix A of the possible Kac-Moody algebras realised in this way. As was the case for $\ell = 2$ theories [3], here too we see that there are several distinct CFT's with different Kac-Moody algebras at each value of the central charge.

Table 3: $\ell = 6$ coset duals for simply laced algebras

|  | $\ell = 0$ | | | $\tilde{\ell} = 6$ | | |
|---|---|---|---|---|---|---|
| No. | $c$ | $h$ | Algebra | $\tilde{c}$ | $\tilde{h}$ | Algebras |
| 2 | 1 | $\frac{1}{4}$ | $A_{1,1}$ | 31 | $\frac{7}{4}$ | A.5 |
| 3 | 2 | $\frac{1}{3}$ | $A_{2,1}$ | 30 | $\frac{5}{3}$ | A.4 |
| 5 | 4 | $\frac{1}{2}$ | $D_{4,1}$ | 28 | $\frac{3}{2}$ | A.3 |
| 7 | 6 | $\frac{2}{3}$ | $E_{6,1}$ | 26 | $\frac{4}{3}$ | A.2 |
| 8 | 7 | $\frac{3}{4}$ | $E_{7,1}$ | 25 | $\frac{5}{4}$ | A.1 |

### An example in detail

Let us illustrate the construction of our $\ell = 6$ two-character CFT's via the above coset construction in some detail using a simple example. Consider a 32-dimensional lattice having the complete root system $A_2^{16}$. The root lattice of $A_2^{16}$ is itself not unimodular, but one can construct an even unimodular lattice which contains this as a sublattice. To do this we need to add in a few vectors from the dual lattice of $A_2^{16}$ such that one obtains a unimodular lattice. This filling (or "gluing") set, as given in [26] is the span of eight vectors which form the rows of the $8 \times 16$ matrix $(I_8, H_8)$, where $I_8$ is the $8 \times 8$ identity matrix and $H_8$ is a certain $8 \times 8$ Hadamard matrix. The resulting lattice is unique, coming from a unique self-dual ternary code. In turn this lattice defines a unique $c = 32$ meromorphic CFT which has $A_{2,1}^{16}$ as its Kac-Moody algebra. The number of spin-1 currents is simply the dimension of the algebra, which is 128. From the $q$-expansion of the single character Eq. (9), we see that $\mathcal{N} = -864$.

We can write the single character of this theory as a non-diagonal modular invariant combination of the affine characters of $A_{2,1}^{16}$. These are of the form $\chi_0^p \chi_1^{16-p}$ where $\chi_0, \chi_1$ are the $A_{2,1}$ characters. They have conformal dimensions in the range $0, \frac{1}{3}, \frac{2}{3}, 1, \cdots, \frac{14}{3}, 5, \frac{16}{3}$. Denote these by $\chi_{m_i}$ where $m_i$ take the above values. The modular invariant (upto a phase) combination of these characters is easily found to be:

$$\chi(\tau) = \chi_0 + 224\chi_2 + 2720\chi_3 + 3360\chi_4 + 256\chi_5 = j(\tau)^{\frac{1}{3}}(j(\tau) - 864). \tag{10}$$

---

[4]This is certainly the case when the lattice is unique for a given root system. The few cases where it is not unique may require additional information.

In fact, the weight enumerator polynomial of the self-dual code constructed from the Hadamard matrix $H_8$ is $(1 + 224y^6 + 2720y^9 + 3360y^{12} + 256y^{15})$. This exemplifies a more general phenomenon: the weight enumerator of a self-dual code determines the Kac-Moody character expansion for the CFT based on the associated lattice [24].

Since this $c = 32$ meromorphic theory has $A_{2,1}$ as one of its direct summands, we can coset it by the $\ell = 0$ two-character $A_{2,1}$ affine theory, to get a new $\ell = 6$ two-character CFT. The bilinear pairing tells us that $c + \tilde{c} = 32$, and $h + \tilde{h} = 2$. Since $c = 2$, and $h = \frac{1}{3}$, we find that the $\ell = 6$ theory has $\tilde{c} = 30$ and conformal dimension $\tilde{h} = \frac{5}{3}$. We can say more, since we know that the new theory has a Kac-Moody algebra $A_{2,1}^{15}$. This determines the integer $N_1$ in line 7 of Table 2 to be 378. Using the known $q$-expansions of the quasi-characters, the characters of our theory are:

$$
\begin{aligned}
\tilde{\chi}_0(\tau) &= (\chi_{A_2}^{n=7})_0 + 378(\chi_{A_2}^{n=1})_0 \\
&= q^{-\frac{5}{4}}(1 + 120q + 109035q^2 + 32870380q^3 + 2612623965q^4 + \cdots), \\
\tilde{\chi}_1(\tau) &= (\chi_{A_2}^{n=7})_1 + 378(\chi_{A_2}^{n=1})_1 \\
&= q^{\frac{5}{12}}(10206 + 5988735q + 669491730q^2 + 32140359765q^3 + \cdots).
\end{aligned}
\tag{11}
$$

These characters can be expressed in terms of the characters of $A_{2,1}^{15}$ as follows. The latter are of the form $\chi_0^p \chi_1^{15-p}$ and have conformal dimensions $0, \frac{1}{3}, \frac{2}{3}, 1, \cdots, \frac{14}{3}, 5$. Analogous to what we did previously, we now label these as $\chi_{m_i}$ where the $m_i$ take the above values (to avoid confusion, we stress that these $\chi_{m_i}$ are not the same as the ones in Eq. (10)). It is then easily verified that:

$$
\begin{aligned}
\tilde{\chi}_0(\tau) &= \chi_0 + 140\chi_2 + 1190\chi_3 + 840\chi_4 + 16\chi_5, \\
\tilde{\chi}_1(\tau) &= 42\chi_{\frac{5}{3}} + 765\chi_{\frac{8}{3}} + 1260\chi_{\frac{11}{3}} + 120\chi_{\frac{14}{3}}.
\end{aligned}
\tag{12}
$$

In this way the coset theory is precisely established as a non-diagonal Kac-Moody invariant.

The $c = 30$ two-character CFT constructed here is unique. However one can construct other two-character $c = 30$ CFT's having different Kac-Moody algebras by starting with a different Kervaire lattice. For example we can find an $\ell = 6$ CFT with algebra $A_{2,1}^{12} \oplus E_{6,1}$ having the same central charge 30 and conformal dimension $\frac{5}{3}$ as the previous one. A list of complete Kac-Moody algebras for $\ell = 6$ two-character CFT's is given in Appendix A. Restricting just to cosets of lattice meromorphic CFT, this is the full list of possible algebras. However there will surely be more general (non-lattice) meromorphic CFT, still having complete Kac-Moody algebras.

# 4 CFT's with an incomplete Kac-Moody algebra

Here we consider meromorphic $c = 32$ CFT with an incomplete Kac-Moody algebra and discuss the possibility of taking their cosets. Because of the difficulty of this problem, our discussion will be briefer and less conclusive than the previous section. Let us first classify the possible types of situations. Leaving out lattice theories with complete root systems, which we have already discussed, the landscape of the remaining even, self-dual lattices is as follows. It includes lattices with root systems of every rank from 0 to 31. Of these, the number of lattices in rank 0 alone is bounded below by $1.096 \times 10^7$ [25]. For rank $1 \le r \le 31$, there is a total of 13,099 distinct root systems. Each one can have a very large number of distinct lattices associated to it.

Given such a daunting number of cases, we cannot carry out a general discussion but will instead try to highlight a few interesting examples. The most extreme example of an incomplete root system is to have no root system at all. A famous lattice with this property is the

Barnes-Wall lattice $BW_{32}$, which has an automorphism group of order $2^{31}.3^5.5^2.7.17.31$. The fact that it has no root system is simply the statement that the minimum (length)$^2$ of any vector in the lattice is greater than 2. Thus, as we have seen, there can be no non-abelian currents, but there are 32 U(1) currents of the form $\partial X^i, i = 1, 2, \cdots 32$. Because it has a very large automorphism group, this lattice can be thought of as a close analogue of the 24-dimensional Leech lattice, whose automorphisms form the Conway group $Co_0$ of order $2^{22}.3^9.5^4.7^2.11.13.23$. Moreover, there is an orbifold of the CFT based on $BW_{32}$ that removes even the abelian current algebra, and the resulting VOA has a larger automorphism group studied in [27]. We may think of this as being analogous in some ways to the Monster CFT at $c = 24$. Following the mathematical literature we will refer to any CFT (or even admissible character) having no Kac-Moody algebra as being of "OZ type" where OZ stands for "one zero" and denotes that the level-1 degeneracy for the identity character is zero [28]. In this notation, the Monster CFT and the CFT of [27] are meromorphic theories of OZ type.

In [4] the possibility of OZ-type coset pairs was considered. The cases considered there had two or three characters and low values of $\ell$, and the coset pairs combined to give the unique $c = 24$ meromorphic CFT of OZ type, namely the Monster CFT. Unfortunately for the case of two characters the coset dual of the $c = -\frac{22}{5}$ minimal was actually not admissible, indeed it had mostly negative coefficients – thus falling in the category of "type II quasi-character" [15]. However, an admissible example was uncovered in the three-character case: the Baby Monster CFT, dual to the Ising model. Due to the OZ nature of the numerator (Monster) and denominator (Ising), one has no Kac-Moody algebra to help in defining the coset. Nevertheless, the existence of the coset dual as a VOA has been established by other means [29, 30] and consistency of its correlation functions was shown in [31]. Very recently [32] other OZ coset pairs have been found, and the duals have large sporadic groups as their automorphisms.

Encouraged by this, we may wonder if there is an $\ell = 6$ two-character CFT obtained by taking the coset of the $BW_{32}$ orbifold by some $\ell = 0$ CFT of OZ type. Unfortunately this does not work, for the same reason as in [4]. For example, an $\ell = 6$ dual of the Lee-Yang minimal model would have $c = \frac{182}{5}$. But this is not in Table 2, and we have verified that it is a quasi-character of type II. We may instead start with admissible OZ-type $\ell = 6$ characters and look for their $\ell = 0$ duals, for example consider the character in line 1 of Table 2 which has $(c, h) = (\frac{122}{5}, \frac{6}{5})$, and choose $N_1 = 244$, the value that removes the degeneracy of the first state above the identity. The $\ell = 0$ theory that pairs up with it $c = \frac{38}{5}, h = \frac{4}{5}$ which is precisely the $E_{7.5}$ theory, identified in [33] as an intermediate vertex operator algebra (IVOA). However, this latter theory is not OZ, as it has a spin-1 algebra of dimension 190, a number that sits between 133 and 248 (the dimensions of $E_7$ and $E_8$) and famously fills a gap in the Deligne series [34]. We have verified that no OZ coset pairs of two-character theories with $\ell = 0$ and $\ell = 6$ exist. Past experience strongly suggests, however, that such pairs may exist from three characters onwards.

Better examples are found by considering each of the entries in Table 2 and first choosing $N_1$ so that they become of OZ type. As explained above, their $\ell = 0$ coset duals are not of OZ type, in fact they are Deligne series CFT's having a simple level-1 Kac-Moody algebra. This suggests that we look for $c = 32$ meromorphic theories with a simple level-1 Kac-Moody algebra, and coset them by a Deligne series CFT. From Table 1 of [25] we see that there are indeed lattices having $A_1, A_2, D_4, E_6$ as their root systems (but curiously not $E_7$). The CFT on these lattices will have an extra U(1)$^{32-r}$ symmetry. Assuming this can be removed by orbifolding, one would find the right kind of meromorphic CFT such that, when quotiented by a simply-laced CFT in the Deligne series (and excluding $E_7$), we will recover our desired $\ell = 6$ CFT of OZ type.

We have looked at just a few special cases of cosets of meromorphic CFT with incomplete Kac-Moody algebras. We identified a few concrete possible examples but did not give a precise proof of the existence of any of these coset CFT's. It should be possible to construct them using

VOA techniques, as was done for the Baby Monster in [29, 30]. Also many more examples can be found, and we leave this subject for future investigation.

## 5 Discussion

In this work we proposed a procedure to classify $\ell = 6$ two-character CFT's as cosets of meromorphic CFT with central charge 32. We identified a number of cases where the Kac-Moody algebra suffices to define the coset precisely, just like the cases originally discussed in [3]. One important conclusion of our investigation is that there is potentially an enormity of $\ell = 6$ two-character CFT that mirrors the enormous number (more than a billion) of meromorphic CFT at $c = 32$.

Given that the task of completely classifying even, unimodular lattices in 32 dimensions is already considered too daunting by mathematicians, it would seem quite hopeless to try and list out all $\ell = 6$ CFT. This is true even before considering orbifolds of lattice theories and other constructions as in [14], which would only expand the list further. Still, it is satisfying to know that two-character CFT's exist in a comparable profusion to meromorphic ones, something that was not previously clear.

Our investigation leaves much to be done. We suggested a way to look for OZ-type theories, and precise constructions of these would be useful. This would involve defining the $c = 32$ orbifold CFT associated to specific lattices that we described in the previous section. It would be nice to construct at least one $\ell = 6$ CFT with a complete but non-simply-laced Kac-Moody algebra. One may want to look at lattices with a root system of rank 31, the closest to being complete, and see if coset VOA's can be defined. Finally one should try to understand the landscape of $\ell > 6$ two-character theories, as well as the barely explored world of higher-character theories with $\ell > 0$.

## Acknowledgements

RC acknowledges the support of an INSPIRE Scholarship for Higher Education, Government of India. We are both grateful for support from a grant by Precision Wires India Ltd, for String Theory and Quantum Gravity research at IISER Pune.

# A List of possible complete Kac-Moody algebras for $\ell = 6$

Below are lists of complete Kac-Moody algebras of $\ell = 6$ two-character CFT's. All the algebras are at level 1.

## A.1 $c = 25$, $h = \frac{7}{4}$

| No. | $\mathcal{N}$ | Kac-Moody algebra |
|-----|-----|-----|
| 1 | 224 | $A_1{}^{21}D_4$ |
| 2 | 272 | $A_1{}^9 D_4{}^4$ |
| 3 | 320 | $A_1{}^3 D_4{}^4 D_6$ |
| 4 | 344 | $A_5{}^3 D_4 E_6$ |
| 5 | 368 | $A_1{}^3 D_4 D_6{}^3$ |
| 6 | 368 | $A_1 A_3 A_7{}^2 D_7$ |
| 7 | 392 | $A_3 A_5 A_{11} D_6$ |
| 8 | 416 | $A_1 D_4 D_6{}^2 D_8$ |
| 9 | 464 | $A_2 A_9 A_{14}$ |
| 10 | 464 | $A_5 A_{11} D_9$ |
| 11 | 464 | $A_9{}^2 E_7$ |

| No. | $\mathcal{N}$ | Kac-Moody algebra |
|-----|-----|-----|
| 12 | 536 | $A_8 A_{17}$ |
| 13 | 464 | $D_6{}^3 E_7$ |
| 14 | 536 | $A_3 A_{15} E_7$ |
| 15 | 512 | $A_1{}^2 D_8{}^2 E_7$ |
| 16 | 512 | $A_1 D_6 D_8 D_{10}$ |
| 17 | 716 | $A_2 A_{23}$ |
| 18 | 608 | $D_6 D_{12} E_7$ |
| 19 | 560 | $D_4 E_7{}^3$ |
| 20 | 704 | $A_1 D_{10} D_{14}$ |
| 21 | 896 | $D_{18} E_7$ |

## A.2 $c = 26$, $h = \frac{4}{3}$

| No. | $\mathcal{N}$ | Kac-Moody algebra |
|-----|-----|-----|
| 1 | 236 | $A_2{}^{10} E_6$ |
| 2 | 326 | $A_2 A_8{}^3$ |
| 3 | 344 | $A_5{}^3 D_4 E_7$ |
| 4 | 416 | $A_5{}^2 D_{10} E_6$ |
| 5 | 512 | $A_3 A_{11} D_{12}$ |
| 6 | 476 | $A_{11} A_{15}$ |
| 7 | 566 | $A_6 A_{20}$ |
| 8 | 656 | $A_{11} D_{15}$ |
| 9 | 806 | $A_{26}$ |

### A.3 $c = 28, h = \frac{3}{2}$

| No. | $\mathcal{N}$ | Kac-Moody algebra |
|---|---|---|
| 1 | 112 | $A_1{}^{28}$ |
| 2 | 128 | $A_1{}^{24}D_4$ |
| 3 | 160 | $A_1{}^{22}D_6$ |
| 4 | 144 | $A_1{}^{20}D_4{}^2$ |
| 5 | 224 | $A_1{}^{21}E_7$ |
| 6 | 176 | $A_1{}^{18}D_4D_6$ |
| 7 | 160 | $A_1{}^{16}D_4{}^3$ |
| 8 | 224 | $A_1{}^{16}D_4D_8$ |
| 9 | 192 | $A_1{}^{14}D_4{}^2D_6$ |
| 10 | 176 | $A_1{}^{12}D_4{}^4$ |
| 11 | 176 | $A_3{}^8D_4$ |
| 12 | 192 | $A_1{}^8D_4{}^5$ |
| 13 | 224 | $A_1{}^{12}D_4D_6{}^2$ |
| 14 | 208 | $A_1{}^{10}D_4{}^3D_6$ |
| 15 | 200 | $A_2{}^4A_5{}^4$ |
| 16 | 272 | $A_1{}^9D_4{}^3E_7$ |
| 17 | 256 | $A_1{}^8D_4{}^3D_8$ |
| 18 | 240 | $A_1{}^8D_4{}^2D_6{}^2$ |
| 19 | 224 | $A_1{}^6D_4{}^4D_6$ |
| 20 | 224 | $D_4{}^8$ |

| No. | $\mathcal{N}$ | Kac-Moody algebra |
|---|---|---|
| 21 | 320 | $A_1{}^6D_4{}^4D_{10}$ |
| 22 | 288 | $A_1{}^6D_4{}^3D_6D_8$ |
| 23 | 272 | $A_1{}^6D_4{}^2D_6{}^3$ |
| 24 | 256 | $A_1{}^4D_4{}^4D_6{}^2$ |
| 25 | 320 | $A_1{}^4D_4{}^2D_6{}^2D_8$ |
| 26 | 304 | $A_1{}^4D_4D_6{}^4$ |
| 27 | 320 | $A_1{}^3D_4{}^4D_6E_7$ |
| 28 | 288 | $A_1{}^2D_4{}^3D_6{}^3$ |
| 29 | 344 | $A_5{}^3D_4E_6E_7$ |
| 30 | 416 | $D_4{}^5D_{12}$ |
| 31 | 352 | $D_4{}^4D_8{}^2$ |
| 32 | 320 | $D_4{}^2D_6{}^4$ |
| 33 | 368 | $A_1{}^3D_4D_6{}^3E_7$ |
| 34 | 384 | $A_1{}^2D_4{}^2D_6{}^2D_{10}$ |
| 35 | 352 | $A_1{}^2D_4D_6{}^3D_8$ |
| 36 | 416 | $D_4{}^2D_8{}^3$ |
| 37 | 416 | $A_1D_4D_6{}^2D_8E_7$ |
| 38 | 544 | $D_4D_8{}^2D_{12}$ |
| 39 | 560 | $D_4E_7{}^4$ |

### A.4 $c = 30, h = \frac{5}{3}$

| No. | $\mathcal{N}$ | Kac-Moody algebra |
|---|---|---|
| 1 | 128 | $A_2{}^{15}$ |
| 2 | 182 | $A_2{}^{12}E_6$ |
| 3 | 236 | $A_2{}^9E_6{}^2$ |
| 4 | 200 | $A_2{}^3A_5{}^4D_4$ |
| 5 | 332 | $A_2A_3{}^2A_{11}{}^2$ |
| 6 | 344 | $A_2{}^3E_6{}^4$ |

| No. | $\mathcal{N}$ | Kac-Moody algebra |
|---|---|---|
| 7 | 326 | $A_8{}^3E_6$ |
| 8 | 368 | $A_2A_{11}{}^2D_6$ |
| 9 | 446 | $A_5A_8A_{17}$ |
| 10 | 464 | $A_2A_{14}{}^2$ |
| 11 | 464 | $A_9A_{14}E_7$ |
| 12 | 716 | $A_{23}E_7$ |

## A.5 $c = 31, h = \frac{5}{4}$

| No. | $\mathcal{N}$ | Kac-Moody algebra |
|-----|------|-------------------|
| 1 | 96 | $A_1{}^{31}$ |
| 2 | 112 | $A_1{}^{27}D_4$ |
| 3 | 144 | $A_1{}^{25}D_6$ |
| 4 | 128 | $A_1{}^{23}D_4{}^2$ |
| 5 | 192 | $A_1{}^{23}D_8$ |
| 6 | 160 | $A_1{}^{21}D_4D_6$ |
| 7 | 144 | $A_1{}^{19}D_4{}^3$ |
| 8 | 144 | $A_1{}^7A_3{}^8$ |
| 9 | 256 | $A_1{}^{21}D_{10}$ |
| 10 | 224 | $A_1{}^{20}D_4E_7$ |
| 11 | 192 | $A_1{}^{19}D_6{}^2$ |
| 12 | 176 | $A_1{}^{17}D_4{}^2D_6$ |
| 13 | 160 | $A_1{}^{15}D_4{}^4$ |
| 14 | 224 | $A_1{}^{15}D_4{}^2D_8$ |
| 15 | 192 | $A_1{}^{13}D_4{}^3D_6$ |
| 16 | 176 | $A_1{}^{12}D_4{}^5$ |
| 17 | 192 | $A_1{}^7D_4{}^6$ |
| 18 | 224 | $A_1{}^{11}D_4{}^2D_6{}^2$ |
| 19 | 208 | $A_1{}^9D_4{}^4D_6$ |
| 20 | 272 | $A_1{}^8D_4{}^4E_7$ |
| 21 | 256 | $A_1{}^7D_4{}^4D_8$ |
| 22 | 240 | $A_1{}^6D_4{}^3D_6{}^2$ |
| 23 | 224 | $A_1{}^5D_4{}^5D_6$ |
| 24 | 272 | $A_1{}^3A_5{}^4D_8$ |
| 25 | 288 | $A_1{}^7D_6{}^4$ |
| 26 | 320 | $A_1{}^5D_4{}^4D_{10}$ |
| 27 | 288 | $A_1{}^5D_4{}^3D_6D_8$ |

| No. | $\mathcal{N}$ | Kac-Moody algebra |
|-----|------|-------------------|
| 28 | 272 | $A_1{}^5D_4{}^2D_6{}^3$ |
| 29 | 256 | $A_1{}^3D_4{}^4D_6{}^2$ |
| 30 | 264 | $A_1{}^3A_7{}^4$ |
| 31 | 320 | $A_1{}^3D_4{}^2D_6{}^2D_8$ |
| 32 | 304 | $A_1{}^3D_4D_6{}^4$ |
| 33 | 320 | $A_1{}^2D_4{}^4D_6E_7$ |
| 34 | 288 | $A_1D_4{}^3D_6{}^3$ |
| 35 | 352 | $A_1A_3{}^2A_7{}^2D_{10}$ |
| 36 | 342 | $A_6{}^3A_{13}$ |
| 37 | 384 | $A_1{}^3D_6{}^2D_8{}^2$ |
| 38 | 368 | $A_1{}^2D_4D_6{}^3E_7$ |
| 39 | 336 | $A_1D_6{}^5$ |
| 40 | 384 | $A_1D_4{}^2D_6{}^2D_{10}$ |
| 41 | 352 | $A_1D_4D_6{}^3D_8$ |
| 42 | 368 | $A_3A_7{}^2D_7E_7$ |
| 43 | 416 | $A_5A_{11}D_5D_{10}$ |
| 44 | 480 | $A_1A_9{}^2D_{12}$ |
| 45 | 480 | $A_1D_6{}^3D_{12}$ |
| 46 | 448 | $A_1D_6{}^2D_8D_{10}$ |
| 47 | 516 | $A_1A_{15}{}^2$ |
| 48 | 416 | $D_4D_6{}^2D_8E_7$ |
| 49 | 512 | $A_1D_8{}^2E_7{}^2$ |
| 50 | 576 | $A_1D_{10}{}^3$ |
| 51 | 512 | $D_6D_8D_{10}E_7$ |
| 52 | 704 | $A_{17}D_{14}$ |
| 53 | 704 | $D_{10}D_{14}E_7$ |
| 54 | 1026 | $A_{31}$ |

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
