# Peer review of "Curiosities above c = 24"

_SciPost Physics, doi:SciPost Phys. 6, 053 (2019)_

## Round 2 · Referee Report · Anonymous (Referee 1) · 2019-3-4

Strengths

Clear exposition of results and assumptions
Several interesting examples

Weaknesses

The ultimate goal of this line of research is not clear.

Report

This paper makes the first steps on a new avenue towards an understanding of CFTs beyond c=24. Although it is not clear where this path will lead, for example given the fact that the c=32 theories are probably not enumerable in practice, there are enough interesting results in this work to warrant publication, so that future work can build on these results.

I only have a minor comment: The abbreviation LY (presumably Lee-Yang) in table 1 should be explained. One should not leave the reader guessing.

Requested changes

Define LY

  • validity: high
  • significance: good
  • originality: good
  • clarity: good
  • formatting: excellent
  • grammar: excellent

Author:  Sunil Mukhi  on 2019-04-02  [id 480]

(in reply to Report 1 on 2019-03-04)
Category:
reply to objection

Sorry, we are including an explanation of LY in a revised version.

---

## Round 2 · Referee Report · Anonymous (Referee 2) · 2019-3-25

Strengths

1 - Useful and original approach to the classification of RCFT
2 - The authors' methods might lead to potentially new interesting RCFT with c>24 could be found.

Weaknesses

1 - The article is not completely self-contained: a review of the previous results by the authors (especially the most recent ones) would make it much clearer

Report

The article's goal is to classify CFTs whose chiral algebra has two characters and whose Wronskian has a certain number of zeroes (l/6=1 in the authors' notation). The starting point is a classification (that was done in previous articles by the authors' and by others) of pairs of admissible characters, i.e. such that their Fourier coefficients are non-negative integers. It is proposed that all (or most) pairs of such admissible characters might correspond to RCFTs obtained by taking cosets of some meromorphic CFT of central charge c=32.

The approach is original and innovative. The methods described here might potentially lead to the discovery of several new RCFT at central charge c>24.

Regarding the clarity of the manuscript, my main concern is that the article is heavily based on results obtained in previous recent articles, and that are not really reviewed, explained or motivated in the present manuscript. This makes the article very difficult to understand even for a reader with a good background in CFT. In particular, for this reason, I was not able to verify that all the authors' statements are correct.

I suggest the authors to add a couple of pages (either in an appendix, or even better as an introductory section), where they review the main results in the references (in particular, reference [15]) that they use in this paper. Once this is done (and provided that the derivations in the article are correct), I think that the manuscript deserves to be published.

Requested changes

As explained in my report, the main change that I would suggest is a (brief) introductory section where the main results that are used by the authors are reviewed and explained. Once this is properly done, some of the following comments might be superfluous:

1 - end of page 1: "The case c=24 ... 71 RCFTs" strictly speaking, the fact that there are 71 meromorphic CFTs at c=24 is based on the unproved conjecture that the Monster VOA is the only one without currents. 2- page 2, line 10: " the number of zeroes of the leading Wronskian, denoted l/6..." I guess it is fractional because it is a function defined on an orbifold (which I guess is the upper half-plane divided by SL(2,Z) ). I would suggest the authors to explain explicitly where the Wronskian is defined and their conventions in counting the multiplicities of zeroes. (In the abstract, the authors claim that l is the number of zeroes of the Wronskian, which means that they are using a different convention for the order of a zero. I understand that it is not a good idea to insert too many complications in the abstract, but at least in the main body this point should be clarified). 3- page 2, lines 18 - 21: "Note that ... given c and l" I would help if the authors write these relations explicitly (and maybe explain in more detail where they come from), at least in the case of two characters. 4- page 3, point (iii) and the following lines: is one supposed add (or take linear combinations of) quasi-characters with the same l? Otherwise, the statement that l is augmented by "multiples of 6" would not make much sense. If this is the case, please say this explicitly. In any case, it is not explained why adding quasi-characters should change the central charge and the dimension of the primary in the way that is described here. I suggest the authors to include some more details and some reference (but the reference alone is not enough). 5- page 7, after point (iii): "Thus all even unimodular lattices ... with dimension less than 24" less or equal to 24.

  • validity: good
  • significance: good
  • originality: good
  • clarity: ok
  • formatting: good
  • grammar: excellent

Author:  Sunil Mukhi  on 2019-04-02  [id 481]

(in reply to Report 2 on 2019-03-25)

Thank you for the helpful report. In a revised version, we have added a couple of pages of introductory material in Sections 1 and 2, as requested. We hope this will make the paper more self-contained. We have also addressed all the points listed in your "requested changes".

---

## Round 3 · Referee Report · Anonymous (Referee 2) · 2019-4-12

Report

The authors implemented the changes that I suggested in my previous report. I recommend the latest version of the article for publication.

---

## Round 3 · Author Response

We have revised the article to incorporate the suggestions of both referees and the editor, notably by adding some introductory material and clarifying a few points.

---

## Round 3 · List of Changes

1. Explained that LY stands for "Lee-Yang"
  2. Noted that at c=24, the number 71 of theories is subject to a uniqueness conjecture about the Monster CFT.
  3. Added supplementary material in Sections 1 and 2 which address all the queries/comments of Referee 2 as well as the Editor's request to make the paper more self-contained.

---

## Editorial Decision

published